# The FMS like Tyrosine Kinase 3 (FLT3) Is Overexpressed in a Subgroup of Multiple Myeloma Patients with Inferior Prognosis

**DOI:** 10.3390/cancers12092341

**Published:** 2020-08-19

**Authors:** Normann Steiner, Karin Jöhrer, Selina Plewan, Andrea Brunner-Véber, Georg Göbel, David Nachbaur, Dominik Wolf, Eberhard Gunsilius, Gerold Untergasser

**Affiliations:** 1Department of Internal Medicine V (Hematology and Medical Oncology), Medical University of Innsbruck, Anichstraße 35, A-6020 Innsbruck, Austria; Selina.Plewan@student.i-med.ac.at (S.P.); david.nachbaur@i-med.ac.at (D.N.); dominik.wolf@i-med.ac.at (D.W.); eberhard.gunsilius@i-med.ac.at (E.G.); gerold.untergasser@i-med.ac.at (G.U.); 2Tyrolean Cancer Research Institute, Innrain 66, A-6020 Innsbruck, Austria; Karin.Joehrer@tkfi.at; 3Department of Pathology, Neuropathology and Molecular Pathology, Medical University of Innsbruck, Müllerstraße 44, A-6020 Innsbruck, Austria; Andrea.Brunner@i-med.ac.at; 4Department of Medical Statistics, Informatics and Health Economics, Medical University of Innsbruck, Schöpfstraße 41/1, A-6020 Innsbruck, Austria; georg.goebel@i-med.ac.at

**Keywords:** multiple myeloma, *FLT3*, midostaurin, gilteritinib, overexpression

## Abstract

Therapy resistance remains a major challenge in the management of multiple myeloma (MM). We evaluated the expression of FLT3 tyrosine kinase receptor (FLT3, CD135) in myeloma cells as a possible clonal driver. FLT3 expression was analyzed in bone marrow biopsies of patients with monoclonal gammopathy of undetermined significance or smoldering myeloma (MGUS, SMM), newly diagnosed MM (NDMM), and relapsed/refractory MM (RRMM) by immunohistochemistry (IHC). *FLT3* gene expression was analyzed by RNA sequencing (RNAseq) and real-time PCR (rt-PCR). Anti-myeloma activity of FLT3 inhibitors (midostaurin, gilteritinib) was tested in vitro on MM cell lines and primary MM cells by ^3^H-tymidine incorporation assays or flow cytometry. Semi-quantitative expression analysis applying a staining score (FLT3 expression IHC-score, FES, range 1–6) revealed that a high FES (>3) was associated with a significantly shorter progression-free survival (PFS) in NDMM and RRMM patients (*p* = 0.04). RNAseq and real-time PCR confirmed the expression of *FLT3* in CD138-purified MM samples. The functional relevance of FLT3 expression was corroborated by demonstrating the in vitro anti-myeloma activity of FLT3 inhibitors on FLT3-positive MM cell lines and primary MM cells. FLT3 inhibitors might offer a new targeted therapy approach in a subgroup of MM patients displaying aberrant FLT3 signaling.

## 1. Introduction

The therapy spectrum for multiple myeloma (MM) has considerably broadened over the past decade, which has led to an improved overall survival (OS), particularly in patients up to 80 years of age [1,2,3]. In the case of unsuccessful first-line therapy, there is a wide range of alternative drugs available, and the former “watch and wait” approach has been replaced by early treatment initiation [4]. The shift from a disease defined by symptoms to a disease defined by biomarkers enables diagnosis and treatment of MM prior to the occurrence of end-organ damage [4,5,6]. Nevertheless, therapy resistance due to genetic alterations, multifocal tumor growth, and subclonal structure remains a big challenge in the treatment of MM [7]. According to the Revised International Staging System (RISS), patients with translocation t(4;14), translocation t(14;16), and deletion del(17p), amongst others, are considered to be high-risk patients with reduced OS and progression-free survival (PFS) [8]. In order to provide patients with personalized therapy regimens, it is an unmet medical need to find additional biomarkers for improved patient stratification and prediction of therapy response as well as new druggable targets for therapeutic intervention [9,10].

In addition to the genetic alterations in MM cells, the respective bone marrow microenvironment supports tumor progression and development of therapy resistances [11,12]. Assessing the prognostic value of several angiogenic factors, our group found that FMS-like tyrosine kinase 3 ligand (FLT3L) levels in peripheral blood and bone marrow plasma correlate directly with the stage of disease [13]. Kokonozaki et al. detected relations between FLT3L serum levels and those of multiple other MM proliferation markers (B-cell activating factor, c-reactive protein, lactate dehydrogenase (LDH), interleukin–6, and bone marrow ki-67 proliferative index) as well as known soluble angiogenic factors, such as vascular endothelial growth factor, tumor necrosis factor-α, endoglin, and hematopoietic growth factor [14,15]. FLT3L activates FMS-like tyrosine kinase 3 (FLT3), a tyrosine kinase receptor type 3, which is strongly expressed in common lymphoid progenitors. The phosphorylation cascades initiated by the FLT3L/FLT3–complex in association with other cytokines, such as interleukin–7 and interleukin–11, have been proven to play an important role in the differentiation of B–cell progenitor cells [16]. Murine experiments showed that an inadequate expression of FLT3L results in deficiencies in hematopoietic progenitors, natural killer cells, and dendritic cells, whilst mice with a null-mutation of the FLT3 gene displayed deficiencies in primitive B–lymphoid progenitors [17,18].

In order to properly understand the role of the FLT3L/FLT3-complex in the course of MM, we analyzed FLT3 expression in bone marrow biopsies of patients with plasma cell dyscrasias (Table 1) by immunohistochemistry and real-time PCR and tested effects of FLT3 inhibitors on MM cell lines and on primary myeloma cells.

Table 1 represents the patient characteristics at the time of sample collection. For most patients of the newly diagnosed multiple myeloma (NDMM) group, the bone marrow samples that were used in this study were the ones that led to diagnosis. Thus, these samples were obtained before the first treatment cycle was started and the respective patients are listed amongst “therapy not started”. The patients in the monoclonal gammopathy of undetermined significance/smoldering myeloma (MGUS/SMM) group did not receive therapy due to their premalignant condition. The category “therapy at the time of sample” solely refers to ongoing therapy at the time of sample collection. Three of the relapsed/refractory multiple myeloma (RRMM) patients had previously received treatment but had not yet been started on a new therapy regimen at the time of sample collection.

## 2. Results

### 2.1. FLT3 Protein Is Expressed in Bone Marrow Samples of MGUS, NDMM, and RRMM Patients

FLT3 tyrosine kinase receptor protein (CD135) was analyzed in an acute myeloid leukemia (AML) patient with FLT3 ITD mutation displaying clusters of positive AML cells within adipocytes and stromal tissue (Figure 1A). FLT3 is also expressed in bone marrow biopsies of all groups of MGUS/MM as detected by IHC (Figure 1B–C). Interestingly, plasma cells from MGUS patients and SMM also showed a clear and strong FLT3 staining around the cell nucleus and in the cytoplasm. AML and MM cells were further examined for subcellular localization of FLT3 protein (Figure 2). FLT3 displayed a vesicular staining pattern around nuclei of leucocytes in the tonsil, used as the positive control, as well as in AML and MM cells, suggesting localization within the endoplasmic reticulum (ER) and in vesicles trafficking to the cell membrane. 

### 2.2. High FLT3 Protein (CD135) Expression in Myeloma Cells Correlates with Shorter Progression-Free Survival

FLT3 expression was evaluated by the use of a semi-quantitative immunochemistry (IHC) expression score, FES (range 1–6), taking into account the percentage of FLT3-positive cells and the intensity of FLT3 staining. There was no significant difference between the three patient groups in regard to FLT3 expression levels according to the FES (Figure 3A). Due to the small sample size, the data from the MGUS/SMM patients (*n* = 3) were excluded from further statistical analyses. As we wanted to focus our investigations on a survival analysis, we also excluded seven patients for whom we were not able to collect substantial follow-up data. The remaining NDMM and RRMM patients (*n* = 32) were grouped for FES > 3 (high, *n* = 15) and FES ≤ 3 (low, *n* = 17). These patient characteristics are listed in Table 2. Comparing the survival of the two groups in a Kaplan-Meier analysis, we found a significantly shorter PFS in the patients with a FES > 3 in direct comparison to those with a FES ≤ 3 (*p* = 0.04; Figure 3B). However, we found no significant difference in OS between the two groups (*p* = 0.296, data not shown).

Table 2 represents the patient characteristics at the time of sample collection. For most patients of the NDMM group, the bone marrow samples that were used in this study were the ones that led to diagnosis. Thus, these samples were obtained before the first treatment cycle was started and the respective patients are listed amongst “therapy not started”. The category “therapy at the time of sample” solely refers to ongoing therapy at the time of sample collection. Three of the RRMM patients had previously received treatment but had not yet been started on a new therapy regimen at the time of sample collection.

### 2.3. High FLT3 Expression Scores in Bone Marrow Biopsies Correlate with Increased ß_2_-Microglobulin Levels

As mentioned above, the NDMM and RRMM patients who were used in our survival analyses were grouped for FES ≤ 3 (*n* = 17) and FES > 3 (*n* = 15). Patient characteristics are listed in Table 2. The median age was 69 years (range 41–85 years) in the FES ≤ 3 group and 71 years (range 49–88 years) in the FES > 3 group.

As already mentioned, patients with a FES > 3 had a shorter PFS than patients with a FES ≤ 3 (*p* = 0.04; Table 3). Additionally, the median serum ß_2_-microglobulin-level was significantly higher in patients with a FES > 3 than in patients with a FES ≤ 3 (*p* = 0.005; Table 3). Increased levels of serum ß_2_-microglobulin in patients with MM are known to be associated with a poor prognosis. Moreover, patients with a FES > 3 trended towards a higher ISS stage (*p* = 0.07; Table 3). There were no significant differences between the groups regarding sex, stage of disease, type of heavy chain or light chain, cytogenetic risk, decreased platelet count, decreased hemoglobin, serum LDH, serum creatinine, serum calcium, bone lesions, therapy lines, or therapy regimen.

### 2.4. FLT3 Tyrosine Kinase Receptor Gene Expression Is Elevated in a Subgroup of MM Patients

After our antibody-based FLT3 protein analysis in bone marrow biopsies, we wanted to confirm FLT3 gene expression. Therefore, we isolated CD138 (syndecan-1, SDC1)+ cells from bone marrow aspirates of refractory MM patients (*n* = 4) and performed RNA sequencing (RNAseq). Data were compared to RNAseq data from blood mononuclear cells obtained from healthy donors (PBMNCs, *n* = 4).

Regarding the tyrosine receptor gene expression pattern of MM cells, we found expression of FLT3 in all samples analyzed. When we normalized samples to transcript number/million (TPM), FLT3 gene expression levels were higher than FLT1, but lower than for the Src family of tyrosine kinases BLK (B lymphocyte kinase), LYN, and HCK. Moreover, FLT3 gene expression was lower than FGFR3 (fibroblast growth factor receptor 3), BTK (Bruton’s tyrosine kinase), JAK1-3 (Janus kinase 1-3), and the highly expressed serine-threonine kinase AKT1 (protein kinase B) (Figure 4A). Compared to PBMNCs from healthy donors, MM cells from RRMM patients displayed higher expression levels of FLT3, FGFR3, and AKT1. Purity of CD138^+-^MM cells was proven by checking SDC1 (CD138) gene expression and a set of established myeloma-marker/differentiation factors, such as MZB1 (marginal zone B and B1 cell-specific protein), XBP1 (x-box binding protein 1), IRF4 (interferon regulatory factor 4, MUM-1), SLAM7 (activating transcription factor 4), and TNF receptor superfamily member 17 (BCMA) in direct comparison to PBMNCs of healthy donors (Figure 4B).

Based on our RNAseq findings on the small group of RRMM, we performed a more detailed FLT3 gene expression analysis on CD138^+^-purified MM cells from bone marrow aspirates of a larger collective of NDMM and RRMM patients (*n* = 24). We quantified FLT3 gene expression levels with a hybridization-probe based multiplex real-time PCR and identified a small subgroup of patients with a higher FLT3 gene expression in NDMM and in RRMM (Figure 5A). In six patients, the number of FLT3 copies per 50 ng of RNA exceeded 2000, whilst in most other patients it was below 1000. We found that this group, when compared to the rest of the patient collective, displayed tendencies towards a higher bone marrow plasma cell percentage. Future studies employing larger patient collectives are, however, necessary to confirm this trend. In addition, FLT3 gene expression levels were determined by real-time PCR in myeloma cell lines and AML cell line (Figure 5B). As expected, AML cells exhibited high levels of FLT3 mRNA, followed by the MM cell lines KMS-12-BM and KMS-12-PE. The MM cell line NCI-H929 had very low amounts of FLT3 transcripts.

### 2.5. FLT3 Inhibitors Block Proliferation of MM Cell Lines and Induce Apoptosis of Primary MM Cells

Next, we investigated whether the number of transcripts correlated with the cell line response to FLT3 inhibitors. Human myeloma cell lines (KMS12-BM, KMS12-PE, NCI-H929) and AML cell line MOLM13 were treated with the FLT3 inhibitors midostaurin and gilteritinib, an AKTserine/threonine inhibitor (API-2) in increasing concentrations ranging from 10 to 1000 nM, and the standard MM proteasome-inhibitor (bortezomib, 5–20 nM). Proliferation was measured by 3H-thymidine incorporation.

An inhibition of proliferation was induced by midostaurin and gilteritinib at concentrations ranging from 100 to 1000 nM in AML cells (MOLM-13, Figure 6A). All MM cell lines analyzed were inhibited by midostaurin at 1000 nM, independently of *FLT3* expression levels (Figure 6B–D). In contrast, the more specific FLT3 inhibitor gilteritinib inhibited cell proliferation only in MM cells with higher *FLT3* gene transcription (KMS12-PE and KMS12-BM), not in NCI-H929 displaying low amounts of *FLT3* transcripts. The AKT inhibitor (API-2) was less effective in inhibiting cell proliferation than midostaurin.

Since primary myeloma cells might not proliferate extensively, we chose to assess the impact of the inhibitors on the viability of plasma cells. Therefore, we prepared bone marrow mononuclear cells (by density gradient centrifugation) and treated the malignant plasma cells together with their environmental cells (mononuclear cells) for 24 h with different concentrations of FLT3 inhibitors. The cell fraction containing plasma cells were gated according to CD38 ^high^/CD45 ^low/neg^ staining. From these cells, we assessed apoptotic cells, i.e., cells that were stained by the apoptosis marker efluoro 780 and/or AnnexinV (for gating strategy see Figure 7A). Moreover, we determined *FLT3* gene expression in CD138^+^-purified MM cells from the respective patients (Figure 7B).

FLT3 inhibitors induced apoptosis in a subgroup of patients at low concentrations (Figure 7B). Gilteritinib was acting only on patients MM#257 and MM#258 at 100 and 1000 nM. Patients MM#261 and MM#262 neither showed a response to standard MM therapy (bortezomib) nor to FLT3 inhibitors.

## 3. Discussion

During the past decade, the use of additional biomarkers has become more and more prevalent in the management of MM. As of 2014, the IMWG diagnostic criteria (myeloma defining events) include three biomarkers (clonal bone marrow plasma cell percentage ≥ 60%, FLC ratio ≥ 100, ≥1 focal lesion in MRI studies) in addition to the traditional CRAB criteria (hypercalcemia, renal damage, anemia, bone lesions) [5]. The R-ISS divides myeloma patients into the stages I-III depending on four different biomarkers (serum ß2–microglobulin, serum albumin, chromosomal abnormalities, and serum LDH) [8]. Due to heterogeneous genetic alterations, comorbidities, and other individual characteristics, the response to therapy can be very heterogeneous amongst MM patients. Hitherto, there is no single prognostic biomarker available that reflects directly on disease promoting dynamics in the bone marrow microenvironment.

In a former study, we established that plasma levels of FLT3L in MM patients correlate with the stage of disease [13]. In our current work, we demonstrate that FLT3, the receptor tyrosine kinase that FLT3L binds to, is expressed in MM cells. To the best of our knowledge, this study is the first to confirm the presence of FLT3 in CD138^+^ plasma cells in patients with MGUS/SMM, NDMM, and RRMM by the use of IHC, rt-PCR, and next generation sequencing (NGS). We developed a semi-quantitative IHC FLT3 expression score (FES) that takes into account the extent and intensity of FLT3 staining (range 1–6, score ≤ 3 and >3). Comparing the two patient groups, we found that patients with a FES > 3 had a significantly shorter PFS than those with a FES ≤ 3 (*p* = 0.04). This pertained to patients that were newly diagnosed at the time of sample collection as well as to relapsed/refractory patients.

In recent years, risk stratification, therapy planning, and biomarker research for MM evolved around cytogenetic alterations. According to the R-ISS, patients with translocation t(4;14), translocation t(14;16), and deletion del(17p), amongst others, are considered to be high-risk patients who face a worse OS and PFS [8]. Recently, Laganà et al. created a computational platform for personalized treatment recommendations for RRMM. The authors generated patient profiles based on DNA mutations and RNAseq data. In their mRNA expression findings, they listed the *FLT3* gene amongst “overexpressed actionable deregulated genes” such as *MAPK, FGFR3,* and *PI3K/AKT* [19]. The FLT3 cytoplasmic domain has been shown to associate with the p85 subunit of phosphoinositol-3-kinase (PI3K), monomeric RAS, and phospholipase C–γ, and results in the phosphorylation and hence activation of these proteins [20] and consequently also further downstream pathways of PI3K/AKT and mitogen-activated protein kinase (MAPK) pathways [21,22]. Thus, targeting the PI3K/AKT or BRAF/MEK pathway might also contribute to blocking FLT3 downstream signaling in MM cells. In the clinical setting, data on the treatment of MM with BRAF–MEK inhibitors are emerging, so far confined to the specific setting of extramedullary disease [23]. It will be interesting to investigate if combining FLT3 inhibitors with BRAF/MEK inhibition reveals synergistic therapeutic effects.

FLT3 inhibitors are already in use in the treatment of AML. In approximately 30% of AML patients, the FLT3 receptor is mutated [24]. An internal tandem duplication (ITD), which leads to autonomous FLT3 activity, can be found in approximately 20% of AML patients [22,25]. FLT3 mutations are “actionable” mutations, which can be targeted with FLT3 inhibitors, such as the first generation FLT3 inhibitor midostaurin. Second generation FLT3 inhibitors, such as gilteritinib, are currently in clinical trials [26,27,28]. Although *FLT3* is not mutated in MM, elevated serum levels of FLT3L in RRMM patients [13] might result in increased activation of the downstream phosphorylation cascades, which could potentially be prevented by FLT3 inhibitors. According to this idea, we demonstrate in our study that FLT3 inhibitors (i.e., midostaurin and gilteritinib) exert promising anti-myeloma activity in vitro. The most effective induction of apoptosis in primary MM cells was achieved by high dosages of midostaurin in four out of six patient samples. Interestingly, the two patient samples not affected by midostaurin showed also no response to the standard myeloma drug bortezomib. A limitation of our study was the short culture conditions for MM cells ex vivo from bone marrow aspirates (24 h) that required higher concentrations of FLT3 inhibitors. Basically, MM patients can be treated over a longer time window with lower concentrations of FLT3 inhibitors to achieve growth inhibition or apoptosis of MM cells.

Further limitations of our study are the retrospective character and the small number of MM patients for extensive statistical analysis. Clearly, a further prospective study with a larger patient population is needed to substantiate our findings and carve out the full potential of FLT3 and its ligand in the context of prognosis and targeted therapy of MM.

## 4. Materials and Methods

### 4.1. Patients and Cell Lines

According to the IMWG criteria, patients with MGUS (*n* = 2), SMM (*n* = 1), NDMM (*n* = 26), and RRMM (*n* = 13) were included in the study population (Table 1) [29]. The study was designed and conducted in accordance with national and international guidelines and with the ethical standards of the Declaration of Helsinki. Investigations have been approved by the authors’ institutional review board (1163/2017 and 1220/2018 Innsbruck).

KMS12-PE, KMS12-BM, NCI-H929 MM cells and MOLM-13 AML cells were purchased from DSMZ (German Collection of Microorganisms and Cell Cultures GmbH, Braunschweig, Germany).

### 4.2. Immunohistochemistry of Bone Marrow Biopsies

IHC for the detection of FLT3 was performed on an automated platform (Benchmark^ULTRA^, Ventana Medical Systems, Tucson, AZ, USA) using a polyclonal antibody against FLT3 (CD135, dilution 1:100, SAB 2900166, Sigma–Aldrich, Darmstadt, Germany). Specificity of FLT3 signals in the ER was proven by competitive blocking of signals with a 25-fold molar excess of human recombinant FLT3 protein (Cat: 10445–H08H, Sino–Biological, Beijing, China) in the first antibody incubation step. Three cases of FLT3–mutated AML and human tonsils served as positive controls for FLT3, respectively. In brief, sections were cut at 1.5 µm from formalin-fixed, paraffin–embedded, and decalcified tissue blocks of bone marrow trephine biopsies. After de-paraffinization, slides were heat-pre–treated with cell-conditioning reagent 1 (CC1, Ventana Medical Systems, Tucson, AZ, USA) for antigen retrieval, and primary antibodies against FLT3 were incubated for 32 min at 37 °C. For visualization, the Ultra View DAB Detection Kit (Ventana Medical Systems, Tucson, AZ, USA) was used in accordance with the manufacturer’s recommendation. Finally, slides were washed in distilled water, counterstained with Hematoxylin (12 min) and Bluing Reagent (4 min), dehydrated in descending order of alcohols, cleared in xylene, and cover–slipped with Tissue-Tek mounting medium (Sakura Finetek, Tokjo, Japan).

### 4.3. Scoring System for FLT3 Staining

All routinely prepared slides were re-evaluated by two observers (AB, NS) in regard to diagnosis and plasma cell content. Visual assessment of FLT3 (CD135) expression was then performed by applying a semi-quantitative FLT3 expression score (FES), taking into account the intensity of FLT3 staining of plasma cells (weak staining = 1, strong staining = 2) and the extent of FLT3-positive stained plasma cells/total plasma cells (0–10% FLT3-positive plasma cells = 1, >10–60% = 2, >60% = 3). The final score was calculated by multiplying the extent of stained cells (1–3) with the staining intensity (1–2) and ranged from 1–6. Only cytoplasmic staining was considered positive for FLT3.

### 4.4. Isolation of CD138 (Syndecan–1) Positive Cells from MM Patients

Bone marrow aspirates were subjected to Ficoll-Paque (Sigma Aldrich, St. Louis, MI, USA) density centrifugation to obtain mononuclear cells. Thereafter, MM plasma cells were purified by magnetic-activated cell sorting (MACS) with anti-human CD138 microbeads (Miltenyi Biotec, Bergisch Gladbach, Germany) according to manufacturer’s protocol. Peripheral bool mononuclear cells were obtained from blood samples of healthy donors after Ficoll–Paque (Sigma Aldrich, St. Louis, MI, USA) density centrifugation.

### 4.5. Subcellular Localization Studies of FLT3 in MM Cells

Plasma cells were selected by CD138+ magnetic beads, cytospinned (Cytospin 3, Shandon, Runcorn, UK), and subsequently analyzed by IHC with anti-FLT3 antibodies. AML cells (MOLM-13) served as the positive control. Air-drying and fixation in cold methanol/acetone were followed by acetone/methanol permeabilization and overnight incubation with primary anti-human anti-FLT3 antibodies. Goat anti-rabbit IgG were utilized for further processing and thereafter, secondary antibody staining, counter staining, and fixation were performed. Slides were imaged on the virtual slide system VS120 (Olympus, Tokjo, Japan).

### 4.6. DNA Extraction and Next Generation Sequencing (NGS) for FLT3 Gene Mutations

Genomic DNA was extracted from CD138–purified MM cells with the Magnapur compact nucleic acid isolation station and the nucleic acid extraction kit (provided by the manufacturer Roche Diagnostics, Mannheim, Germany). Concentration of isolated DNA was determined by Qubit 3.0 Fluorometer (Thermo Fisher Scientific, Waltham, MA, USA) and 30 ng of genomic DNA were used to generate libraries for NGS analysis. Prior to multiplexing and cluster generation, the quantity of all NGS libraries was assessed by the Agilent Bioanalyzer 2100 system and the high sensitivity DNA kit (both Agilent Technologies, Santa Clara, CA, USA). Finally, paired-end sequencing was performed with the Miseq Reagent Kit V2 on the Miseq NGS machine (both Illumina, San Diego, CA, USA). Fast files were processed and variants were called by the use of TruSeq Amplicon program (BaseSpace Workflow), 2.0.0.0 Isis (Analysis Software), 2.6.21.7 SAMtools 1.2, Isis Smith–Waterman–Gotoh (Sequence Alignment), 2.6.21.7 Somatic Variant Caller 4.0.13.1, and IONA (Annotation Service) 1.0.10.37.

### 4.7. RNA Extraction and Quantitative Multiplex Real-Time Analysis of FLT3 Gene Expression

For total RNA isolation of bone marrow CD138+ myeloma cells, the RNeasy Mini kit (Qiagen, Hilden, Germany) was used according to the manufacturer’s instructions. Thereafter, genomic DNA in the RNA samples was digested with the DNase I (New England Biolabs, Ipswich, MA, USA). The cDNA was amplified from 1 μg of total RNA using the SuperScript II Reverse Transcriptase Kit (Invitrogen Life Technologies, Waltham, MA, USA). For validation, quantitative real-time polymerase chain reaction (RT-PCR) was performed using sets of gene-specific primers and hybridization probes (Eurofins Genomics, Ebersberg, Germany) on the iQ5 real-time detection system (BioRad, Hercules, CA, USA), *Flt3* Probe: FAM-5-TCCAAATTCCAGCATGCCTGGTTCAAG-BHQ1; *Flt3* For: 5-TTTCACAGGACTTGGACAGAGATTT; *Flt3* Rev: 5-GAGTCCGGGTGTA-TCTGAACTTCT; *ACTB* TexRed-5-TTCACCACCACGGCCGAGC-3-TQ3; *ACTB* For: 5-TGACGGGGTCACCCACA; *ACTB* Rev: 5-CTAGAAGCATTTGCGGTGGA. For absolute quantification, the external plasmid with the *FLT3* coding sequence (NM_004119) was used.

### 4.8. 3′-RNA Whole Transcriptome Sequencing (WTS) of CD-138^+^-Purified MM Cells

One µg of total RNA of CD138+-purified myeloma cells or PBMNCs was prepared for Massive Analysis of cDNA Ends (MACE) with the library preparation kit (v2.0) of GenXPro (Frankfurt, Germany), making use of digital error corrections with unique molecular identifiers. In brief, for each sample, first-strand and second-strand synthesis were performed in a pre-pooling step. Then, 50 ng of each indexed cDNA samples were pooled, fragmented with the Bioruptor Plus (Diagenode, Denville, NY, USA), end-repaired, ligated, and PCR–amplified. Libraries were checked for fragment size (150–200 bp) with the Agilent Bioanalyzer 2100 system with the highly sensitive DNA kit (Santa Clara, CA, USA). Then, 1.5 pM of the pooled library were sequenced with the Nextseq 500/550 high Output Kit v2.5 (75 cycles, 400 M reads) on the Nextseq 500 (Illumina, San Diego, CA, USA) using 20% PhiX library (Illumina, San Diego, CA, USA). Raw sequencing data of 3′ ends were demultiplexed (BCl2-Fastq converter, Illumina, San Diego, CA, USA), PCR duplicates removed (UMI error correction), adaptor and quality trimmed, mapped to reference sequence (Bowtie–2), annotated, and quantified to sequencing depth, i.e., transcripts/million reads = TPM (CCL–Bio, Biomedical Genomics Workbench 5.0, Qiagen, Hilden, Germany).

### 4.9. 3. H-thymidine Proliferation Assay

Proliferation was measured by ^3^H-thymidine incorporation assays as previously described by Jöhrer et al. [30]. Briefly, myeloma and AML cell lines (5  ×  10^4^ cells/well), respectively, were seeded into a 96-well plate with/without the addition of FLT3 inhibitors as described and 1 μCi of ^3^H-thymidine (Perkin Elmer, Waltham, MA, USA) was added to each well for the whole incubation period (48 h for KMS12-PE, KMS12B-M, and MOLM 13; 72 h for NCI-H929). Midostaurin (PKC412) was purchased from Selleckchem (Huston, TX, USA), Gilteritinib (ASP2215) from Medchemexpress (Monmouth Junction, NJ, USA), and bortezomib from Selleckchem (Huston, TX, USA), and compounds were dissolved in DMSO (Sigma Aldrich, St. Louis, MI, USA) to stock solutions of 10 mM. Cells were harvested onto glass fiber filters, scintillation liquid was added, and radioactivity was measured using a scintillation counter (Beckman Coulter, model LS-6500, Brea, CA, USA). Experiments were repeated at least three times in triplicates and statistical analyses were performed utilizing the paired Mann-Whitney U test.

### 4.10. Apoptosis Assay

Patients’ bone marrow aspirates were subjected to Ficoll separation and the mononuclear fraction was collected. For treatment with FLT3 inhibitors, 1 × 10^5^ cells/well were seeded into a 96-well plate and incubated with or without inhibitors at 1000 nM, 100 nM, and 10 nM, respectively, for 24 h. Apoptosis was assessed by Annexin V/efluoro 780 staining (Invitrogen, Thermo Fisher Scientific, Waltham, MA, USA) of CD38 ^high^/CD45 ^neg-low^-stained cells and analysis was performed by flow cytometry (FACS Canto II, BD Biosciences, Franklin Lakes, NJ, USA). The extent of apoptosis was calculated as the percentage of Annexin Vpos plus AnnexinVpos/efluoro 780 pos cells, respectively. All experiments were performed in triplicates and appropriate solvent controls were always included. The standard myeloma drug bortezomib was included as a reference treatment (24 h, 20 nM/10 nM).

### 4.11. Statistical Analysis

We developed a semi-quantitative score, the FLT3 expression score (FES), that combines information about the percentage of plasma cell infiltration and the FLT3 staining intensity [plasma cell infiltration: 0–10% FLT3-positive plasma cells = 1, >10–60% = 2, >60% = 3; staining intensity: weak staining = 1, strong staining = 2]. The final score was calculated by multiplying the extent of staining with the staining intensity and ranged from 1–6. Only cytoplasmic staining was considered positive for FLT3. We chose > 3 as the cut-off for high FLT3 expression, as the median FLT3 expression score for all 32 NDMM/RRMM patients was at 3.

Descriptive data are shown using median (range) and/or mean (standard deviation) for quantitative variables. Absolute/relative frequencies are presented for categorical items. We mainly used non–parametric tests (Mann-Whitney U test, Wilcoxon test, Kruskal-Wallis test) or the Chi² test to test differences between groups. Survival data were illustrated by Kaplan-Meier curves and tested by the log-rank test. The hazard ratio for the PFS was calculated by a univariate Cox-regression model. Proportional hazard assumption was investigated by visual inspection of KM curves. All tests for statistical significance were two-sided. p-values less than 0.05 were considered as statistically significant. Statistical evaluation was performed using SPSS statistical software (version 24.0; SPSS Inc., Chicago, IL, USA).

## 5. Conclusions

A high FLT3 expression score (FES > 3) is associated with a shorter PFS in patients with NDMM and RRMM. FLT3 inhibitors showed activities in a subgroup of MM primary cells and MM cell lines. Thus, FLT3 inhibitors might offer a new targeted therapy approach for a subgroup of patients with ectopic FLT3 signaling.

## Figures and Tables

**Figure 1 cancers-12-02341-f001:**
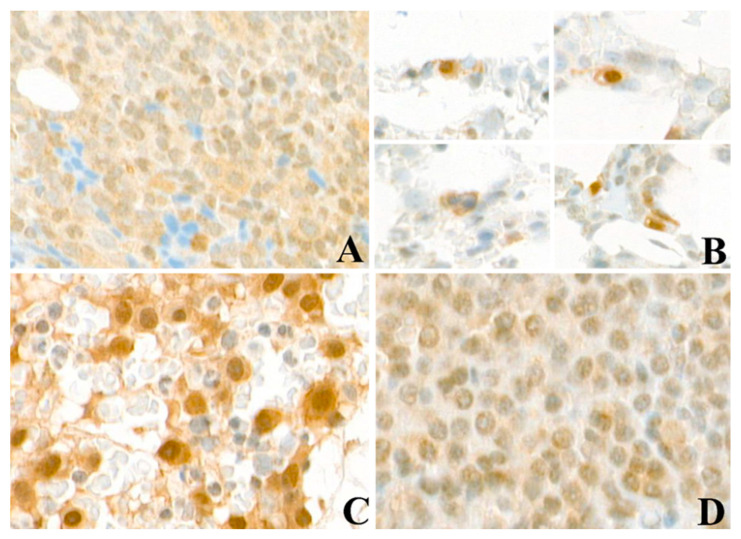
FLT3 expression in bone marrow of acute myeloid leukemia (AML) and multiple myeloma (MM) patients. (**A**) FLT3 expression in a case of AML shows weak cytoplasmic staining in the neoplastic cells. (**B**) Single interstitial plasma cells in a case of monoclonal gammopathy of undetermined significance (MGUS) shows cytoplasmic staining and sometimes exhibit perinuclear vacuoles due to entrapped immunoglobulins. Examples of MM patient samples with strong (**C**) and weak (**D**) cytoplasmic FLT3 staining. Magnification 40×.

**Figure 2 cancers-12-02341-f002:**
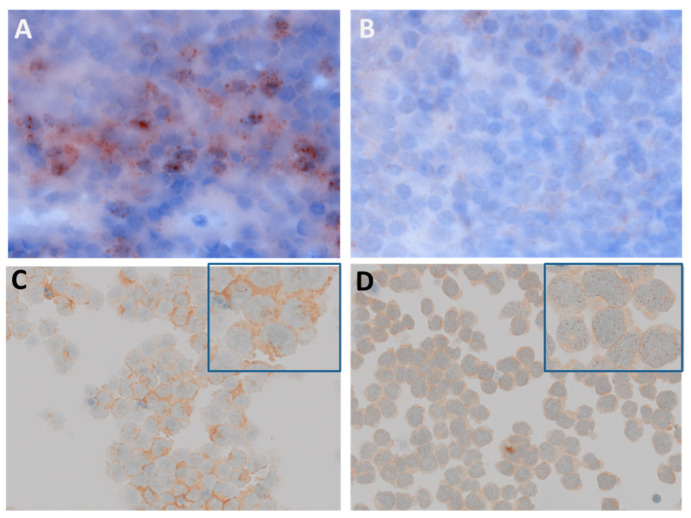
Subcellular localization of FLT3 protein in human tonsil tissue, AML, and MM cells. (**A**) Human tonsil tissue was used as a positive control and displayed leucocytes with ER localization of the FLT3 receptor protein. (**B**) Signals were competitively blocked by adding a 25-fold excess of recombinant FLT3 protein (magnification 100×). (**C)** Cytospin preparations of FLT3-ITD positive AML cells displayed a vesicular expression pattern pointing to ER localization. (**D**) Cytospin preparations of MM cells displayed a vesicular staining pattern around the nucleus and in the cytoplasm (magnification 40×, enlarged image with blue-frame 100×).

**Figure 3 cancers-12-02341-f003:**
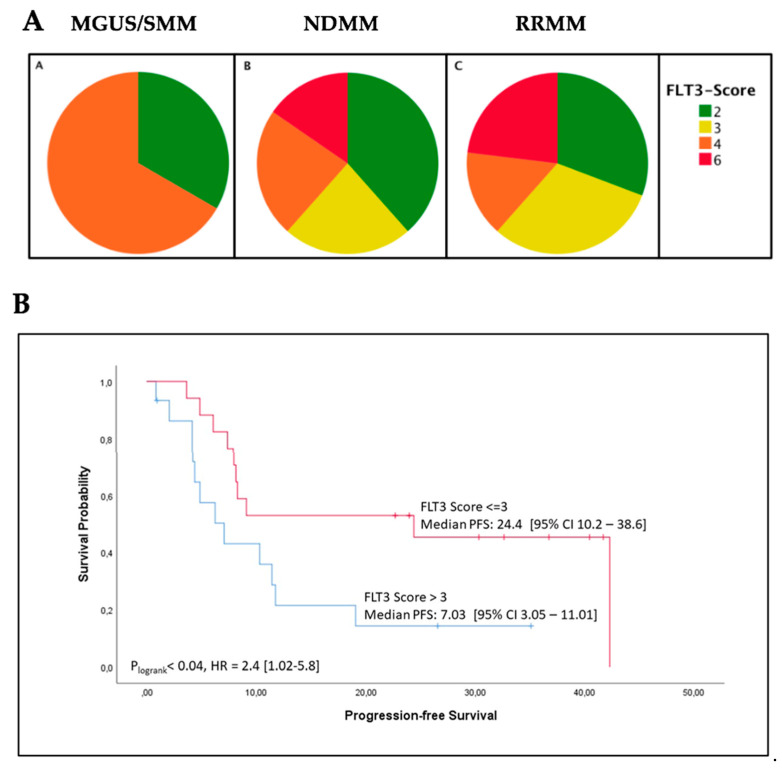
(**A**) Distribution of FLT3 expression immunohistochemistry score (FES) values within the MGUS/SMM (A, *n* = 3), NDMM (B, *n* = 26), and RRMM (C, *n* = 13) patients. The FLT3 expression did not differ significantly between the three patient groups according to the semi-quantitative FLT3 IHC expression score (FES, range 2–6). (**B**) The progression-free survival (PFS) (in months) of MM patients with high FLT3 expression (FES > 3; *n* = 15) was significantly shorter in comparison to patients with low FLT3 expression (FES ≤ 3; *n* = 17); *p* = 0.04, HR = 2.4.

**Figure 4 cancers-12-02341-f004:**
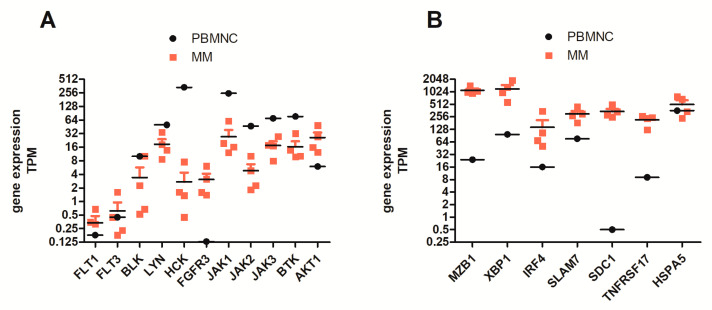
RNA sequencing (RNAseq) data of CD138^+^-purified MM cells of RRMM patients versus peripheral blood mononuclear cells (PBMNCs) of healthy donors. (**A**) PBMNCs (mean of 4 samples) were used for comparison of tyrosine kinase expression levels in RRMM patients (*n* = 4). FLT3 gene expression levels were compared to FLT1, FGFR3, Src-tyrosine kinases BLK (B lymphocyte kinase), LYN and HCK, Janus kinases (JAK1-3), Bruton’s tyrosine kinase (BTK), and the highly expressed serine-threonine kinase AKT1. MM cells displayed higher gene expression of FLT3, FGFR3, and AKT1 than PBMNCs. (**B**) RNAseq data on expression of CD138^+^-purified MM cells versus PBMNCs (mean of 4 samples) of healthy donors. SDC1 (CD138) was expressed at equally high levels in all RRMM patients analyzed, indicating a high purity of MM cells used for transcriptome analysis. Moreover, all samples expressed high levels of B-cell specific transcription factors, such as MZB1, XBP1, IRF4, SLAMF7, TNFRSF17 (BCMA), and Unfolded Protein Response gene HSPA5 as the control. TPM indicates normalized transcripts per million reads analyzed.

**Figure 5 cancers-12-02341-f005:**
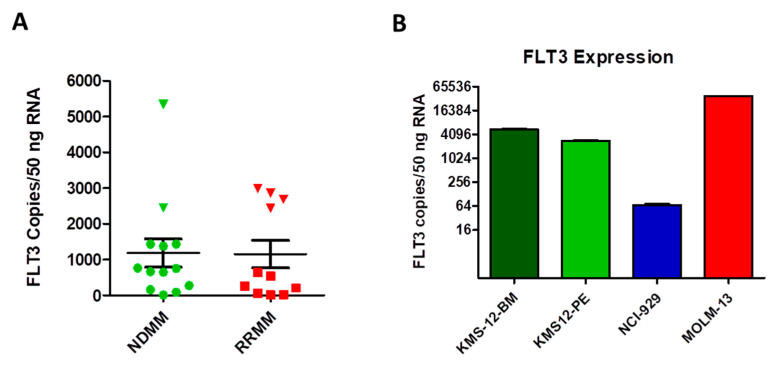
Real-time PCR analysis of FLT3 gene expression in CD138^+^-purified MM cells and MM cell lines. (**A**) CD138^+^ cells were purified from bone marrow aspirates of NDMM and RRMM patients and FLT3 transcript copies were calculated in 50 ng of total RNA input after normalization of samples by Actin beta. Copy numbers were calculated by the use of an external standard consisting of serial dilutions of a FLT3-cDNA-containing plasmid. In both NDMM and RRMM, a subgroup of patients displayed higher FLT3 gene expression (triangles, >2000 copies). (**B**) FLT3 RT-PCR was performed on cDNA of human myeloma cell lines (KMS12-BM, KMS12-PE, NCI-H929), and the AML cell line MOLM13 and FLT3 gene transcripts were quantified as above.

**Figure 6 cancers-12-02341-f006:**
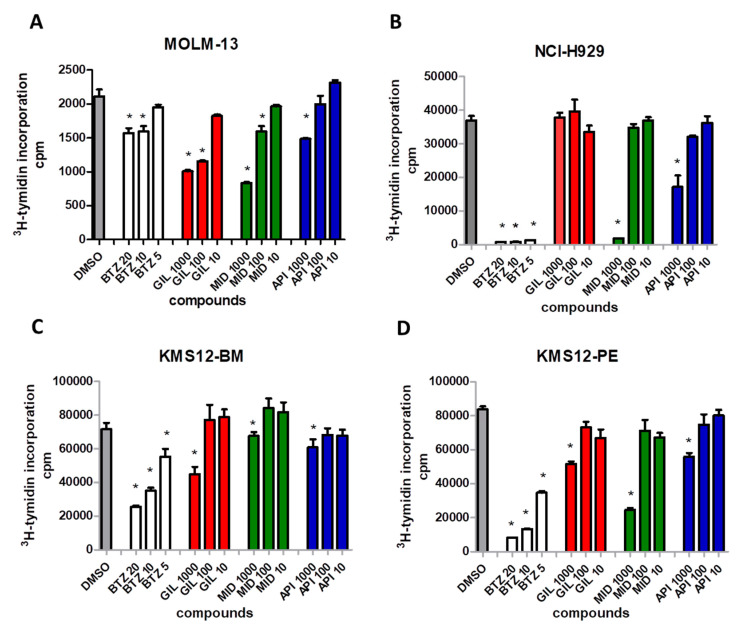
Inhibition of cell proliferation by FLT3 inhibitors in MM and AML cell lines. (**A**) AML cells (MOLM-13) were treated with the FLT3 inhibitors gilteritinib (GIL) and midostaurin (MID), the AKT serine/threonine inhibitor API-2 (API), and the proteasome inhibitor bortezomib (BTZ) in increasing concentrations (10 to 1000 nM, 5–20 nM for bortezomib). (**B**–**D**) MM cell lines with differential expressions of *FLT3* were treated with the FLT3 inhibitors gilteritinib (GIL) and midostaurin (MID), the AKT serine/threonine inhibitor API-2 (API), and the proteasome inhibitor bortezomib (BTZ). Gilteritinib-dependent inhibition of proliferation correlated with *FLT3* expression, whilst effects of midostaurin displayed no correlation with *FLT3* expression status. CPM indicates counts per minute; stars indicate *p* values < 0.05.

**Figure 7 cancers-12-02341-f007:**
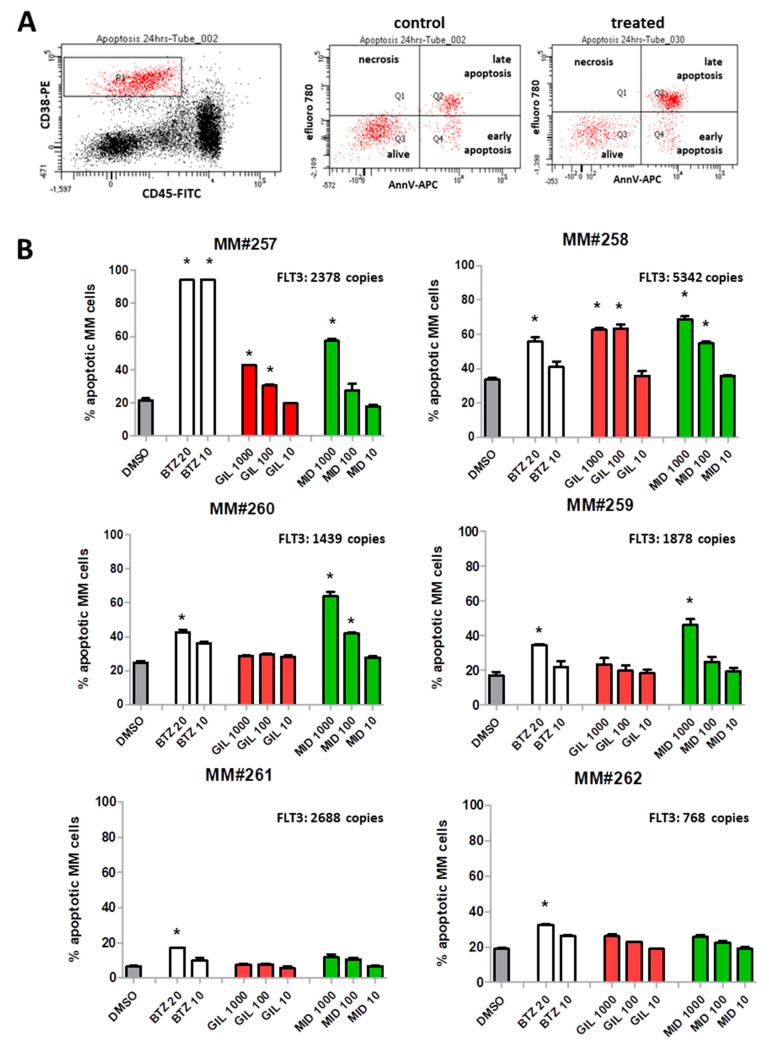
Induction of apoptosis in primary myeloma cells by FLT3 inhibitors. (**A**) Gating strategy for analysis of apoptotic cells: whole bone marrow mononuclear cells were stained with CD38-PE/CD45-F/AnnV-APC/efluoro780. Plasma cells (CD38high/CD45low-median) were gated in P1 (red cells, left dot blot). These cells were then further analyzed for staining with markers AnnexinV and efluoro780. Apoptotic cells are displayed in the lower right (Q4) and upper right (Q2) quadrant. Necrotic cells appear in Q1 and were always <1.5%. MM#258 cells were treated with 0.1% DMSO (control dot blot, middle), or MM#258 cells were treated with 1000 nM gilteritinib (treated dot blot, left). (**B**) Primary myeloma cells were treated with FLT3 inhibitors (midostaurin, MID; gilteritinib, GIL; 10–1000 nM) and standard proteasome-inhibitor bortezomib (BTZ; 5–20 nM). Percentages of apoptotic cells are shown (percentage of cells in Q2 + Q4); * *p* < 0.05; CD138-purified MM cells of all patients were also analyzed for *FLT3* gene expression (FLT: copies displayed within each graph, upper right).

**Table 1 cancers-12-02341-t001:** Patient demographics and characteristics (*n* = 42).

Parameter	MGUS/SMM (*n* = 3)	%	NDMM (*n* = 26)	%	RRMM (*n* = 13)	%
Age, years Median (range)	69 (48–76)		74 (45–88)		66 (41–88)	
Sex						
Female	0	0	12	46.2	6	46.2
Male	3	100	14	53.8	7	53.8
ISS Stage						
No Stage	3	100	1	3.8	0	0
I	0	0	5	19.2	3	23.1
II	0	0	3	11.5	4	30.8
III	0	0	17	65.4	6	46.2
Type of heavy chain (serum)						
IgG	2	66.7	12	46.2	6	46.2
IgM	0	0	0	0	2	15.4
IgA	1	33.3	5	19.2	2	15.4
IgD	0	0	1	3.8	0	0
Light chain only	0	0	8	30.8	3	23.1
Type of light chain (serum)						
Kappa	3	100	16	61.5	8	61.5
Lambda	0	0	9	34.6	5	38.5
Mixed gradient	0	0	1	3.8	0	0
ß2–microglobulin > UNV	1	33.3	22	88	11	84.6
LDH > UNV	1	33.3	8	30.8	4	30.8
Creatinine ≥ 1.3 mg/dL	1	33.3	12	46.2	3	23.1
Serum calcium > UNV	0	0	3	11.5	0	0
Hemoglobin ≤ 12 g/dL	2	66.7	22	84.6	11	84.6
Platelets < 100,000/mm^3^	1	33.3	4	15.4	6	46.2
Osteolytic bone lesions	0	0	22	84.6	13	100
Cytogenetic risk						
Standard risk	1	33.3	3	11.5	1	7.7
High risk	1	33.3	11	42.3	5	38.5
Not available	1	33.3	12	46.2	7	53.8
Therapy lines						
Therapy not started	3	100	22	84.6	0	0
1st line	0	0	4	15.4	4	30.8
2nd line	0	0	0	0	1	7.7
3rd line	0	0	0	0	2	15.4
4th line	0	0	0	0	3	23.1
5th line	0	0	0	0	2	15.4
7th line	0	0	0	0	1	7.7
Therapy at the time of sample						
No active therapy	3	100	22	84.6	3	23.1
PI-based	0	0	1	3.9	3	23.1
IMiD-based	0	0	0	0	5	38.5
PI and IMiD	0	0	3	11.5	1	7.7
Others	0	0	0	0	1	7.7

N—number of patients; ISS—International Staging System; Ig—immunoglobulin; UNV—upper normal value; LDH—lactate dehydrogenase; IMiD—immunomodulatory drugs; PI—proteasome inhibitor.

**Table 2 cancers-12-02341-t002:** Patient characteristics in FES ≤ 3 versus > 3.

Parameter	FES ≤ 3 (*n* = 17)	%	FES > 3 (*n* = 15)	%
Age, years Median (range)	69 (41–85)		71 (49–88)	
Stage of disease				
NDMM	11	64.7	10	66.7
RRMM	6	35.3	5	33.3
Sex				
Female	7	41.2	8	53.3
Male	10	58.8	7	46.7
ISS Stage				
I	7	41.2	1	6.7
II	3	17.6	3	20
III	7	41.2	11	73.3
Type of heavy chain (serum)				
IgG	7	41.2	7	46.7
IgM	1	5.9	1	6.7
IgA	3	17.6	4	26.7
IgD	1	5.9	0	0
Light chain only	5	29.4	3	20
Type of light chain (serum)				
Kappa	8	47.1	11	73.3
Lambda	9	52.9	4	26.7
ß2-microglobulin > UNV	13	76.5	14	93.3
LDH > UNV	3	17.6	3	20
Creatinine ≥ 1.3 mg/dL	5	29.4	7	46.7
Serum calcium > UNV	2	11.8	1	6.7
Hemoglobin ≤ 12 g/dL	13	76.5	13	86.7
Platelets < 100,000/mm^3^	2	11.8	6	40
Osteolytic bone lesions	15	88.2	14	93.3
Cytogenetic risk				
Standard risk	6	35.3	8	53.3
High risk	11	64.7	5	33.3
Not available	0	0	2	13.3
Therapy line				
Therapy not started	9	52.9	8	53.3
1st line	4	23.5	4	26.7
2nd line	1	5.9	0	0
3rd line	0	0	1	6.7
4th line	2	11.8	1	6.7
5th line	1	5.9	0	0
7th line	0	0	1	6.7
Therapy at the time of sample				
No active therapy	11	64.7	9	60
PI-based	1	5.9	3	20
IMiD-based	2	11.8	1	6.7
PI and IMiD	2	11.8	2	13.3
Others	1	5.9	0	0

N—number of patients; FES—FLT3 expression score; ISS—International Staging System; Ig—immunoglobulin; UNV—upper normal value; LDH—lactate dehydrogenase; IMiD—immunomodulatory drugs; PI—proteasome inhibitor.

**Table 3 cancers-12-02341-t003:** Patients with a FES > 3 had a shorter progression-free survival (PFS), a higher ß_2_-microglobulin level, and revealed a trend towards a higher ISS stage in comparison to patients with a FES ≤ 3.

Variable		FES ≤ 3 (*n* = 17)	FES > 3 (*n* = 15)	*p* Value
PFS (months)	Median (95% CI)	24.4 [10.2–38.6]	7.03 [3.05–11.01]	0.04
ß2-microglobulin (mg/L)	Median (95% CI)	4.3 [2.6–10.3]	11.4 [3.7–14.6]	0.005
ISS stage III	N/%	7/41%	11/73%	0.07

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
