# Peer review of "The FMS like Tyrosine Kinase 3 (FLT3) Is Overexpressed in a Subgroup of Multiple Myeloma Patients with Inferior Prognosis"

_cancers, 2020, doi:10.3390/cancers12092341_

Round 1
Reviewer 1 Report
This study discusses the role of FLT3 in multiple myeloma and justifies its potential as a druggable target. While the general idea is interesting, there remain some major caveats in experiment design and data interpretation.
- In Fig.2, the author provided evidence to support the localization of FLT3 in ER and plasma. However, in Fig. 1, most staining is clearly enriched in nuclear. Please explain this inconsistency.
- The data presentation and interpretation in Fig. 4 is confusing and misleading. In Fig. 4A, the author concluded that samples expressed high levels of indicated genes. How was the “high expression” defined? In another word, what served as the “low expression” control? In Fig. 4B, the author compared the reads across different genes, which was an uncommon approach to interpret the RNAseq data. A more reasonable analysis is to compare those gene expression in advanced patients and low-stage patients or healthy hosts.
- Fig. 5 is too descriptive and preliminary. The author identified a subgroup of patients with a higher FLT3. However, without further interpretation (like the clinical manifestation of these patients), it is difficult to evaluate the meaning of those data.
- The way to present apoptosis in Fig.7 is indirect and unconvincing. The authors are suggested to measure the apoptosis in more direct approaches such as TUNEL assay.
Author Response
REVIEWER 1
- “In Fig.2, the author provided evidence to support the localization of FLT3 in ER and plasma. However, in Fig. 1, most staining is clearly enriched in nuclear. Please explain this inconsistency.”
Response: We thank the reviewer for this helpful comment and elaborated Figure 1 with higher magnification (40x) and better resolution (1200dpi) of all images to see cytoplasmic/perinuclear ER FLT3 staining and localization in bone marrow samples of AML/MGUS and MM patients and we adapted the figure legend in section 2.1 to the new images. Due to demineralization of bone in the staining procedure of the formalin-fixed paraffin-embedded (FFPE) bone marrow biopsies the staining quality and resolution is not optimal. The images on AML and MM cells in figure 2 were performed on cytospins with thin layers of cells that allow better visualization of subcellular localization of FLT3 staining than FFPE sections.
2. „The data presentation and interpretation in Fig. 4 is confusing and misleading. In Fig. 4A, the author concluded that samples expressed high levels of indicated genes. How was the “high expression” defined? In another word, what served as the “low expression” control? In Fig. 4B, the author compared the reads across different genes, which was an uncommon approach to interpret the RNAseq data. A more reasonable analysis is to compare those gene expression in advanced patients and low-stage patients or healthy hosts.“
Response: We thank the reviewer for this comment and included additional RNAseq data on peripheral blood mononuclear cells (PBMNCs) of healthy controls in the elaborated new figure 4 and legend to figure 4. These data help to see purity of CD138+ selected cells of RRMM samples by the established MM gene marker set and also depict kinase gene expression levels in MM cells in comparison to mononuclear cells of healthy donors. Interestingly, RRMM displayed higher transcription of FLT3, FGFR3 and AKT1 genes than PBMNCs. The respective result section 2.4 has been adapted.
3. „Fig. 5 is too descriptive and preliminary. The author identified a subgroup of patients with a higher FLT3. However, without further interpretation (like the clinical manifestation of these patients), it is difficult to evaluate the meaning of those data.“
Response: Thank you for pointing out that this information needs better embedding into the context of our paper in order to show its relevance to the issue. We have now included a text passage describing how patients with a higher FLT3 expression (> 2000 copies / 50 ng total RNA) showed tendencies towards a higher plasma cell percentage in comparison to individuals with lower FLT3 expression levels in section 2.4. We are aware of the fact that this group of patients is by far too small to make statistically significant points. However, we found it important to mention this finding, as the FLT3 expression levels were considerably lower in the other patients (the copy number was < 1000 in most cases). Future studies of larger patient collectives might be able to carve out how this affects the course of disease in the individuals concerned. In addition, we added also the information of FLT3 copy numbers for the cell lines analyzed.
4. „The way to present apoptosis in Fig.7 is indirect and unconvincing. The authors are suggested to measure the apoptosis in more direct approaches such as TUNEL assay.“
Response: We apologize for the inconvenience that might be caused by presenting our data in this way. In order to present our data more clearly, we now added our gating strategy of the FACS analysis and calculated the percentage of apoptotic cells instead of “viable cells”. Additionally, we added values for RT-PCR analysis of FLT3 expression for each patient. We chose FACS instead of TUNEL assay, because we incubated whole bone marrow mononuclear cells and by specific FACS staining (CD38/CD45) we can gate the malignant plasma cells and individually collect the data on their apoptosis. By using TUNEL assay, we would not be able to distinguish the effects on the different cell types in the sample.
Reviewer 2 Report
The authors checked the FLT3 experession from bone marrow samples of MGUS, NDMM, and RRMM patients showing survival impact. They also used FLT3 inhibitors with myeloma cell lines and primary patient samples showing potential effects of FLT3 inhibitors. Though the manuscript is novel, there are still some questions for this study.
- The patient number is too small for each group, the MGUS, NDMM, RRMM, to got clear conclusion. And in the table 1, for total 25 NDMM patients, only 4 of them had 1st line of therapy, how about other NDMM patients? Did they received the therapy? Again, of all 13 RRMM patients, only 10 had therapy type in table 1. Therefore I think the table 1 should be update for the number mistakes or should have more description about the number discripency.
- From the table 2, the author stated that FLT3 epxression have PFS impact, but there is no patient numbers on each group.
- From Table 2, the authors stated that FES score had impact on PFS. Since the risks of survival is totally different in different subgroups, such as good survival of MGUS, the authors should not take all patients from 3 different groups together to see the impact. They should see the impct of FES score of FLT3 in different groups seperately.
- The same prolem as table 1 in table 2, the number on the table does not match the total patient number in FES low/high groups. Eespecially the number of therapeutic lines and therapy types did not match the total number in each group. Since the theray lines and treatment agents had strong impact on patient survival, the authors should make the number in the table right for a clear picture.
- From figure 5, how about the expression of normal or MGUS samples?
- In figure 6, the FLT3 inhibitors, mido and gel, can inhibit DNA synthesis in myeloma cell lines in very concertrations, how about the FLT3 expression on these cell lines? Could the authors use other myeloma cell lines with low FLT3 expression?
- Some patients samples showed potential effects of FLT3 inhibitors, how about the FLT3 expression in this samples? Does these patients belong to high expression group? If so, how about other patients?
Author Response
REVIEWER 2
1. „The patient number is too small for each group, the MGUS, NDMM, RRMM, to got clear conclusion. And in the table 1, for total 25 NDMM patients, only 4 of them had 1st line of therapy, how about other NDMM patients? Did they receive the therapy? Again, of all 13 RRMM patients, only 10 had therapy type in table 1. Therefore, I think the table 1 should be update for the number mistakes or should have more description about the number discrepancy.
Response: Thank you for drawing our attention to this numerical inconsistency in both, Table 1 and Table 2. Both tables aim to depict the status quo at the time of sample collection. As many of the NDMM bone marrow samples that we used in this study were the ones that led to the initial diagnosis, those patients had not started therapy at the time of sample collection and thus were not assigned to a therapy line. We have now added a separate category called “therapy not started” and inserted a text passage explaining this circumstance. The same pertains to the RRMM patients: all of them had received one or more therapy lines, but some of them had not yet been started on a new therapy regimen at the time of sample collection and were therefore not listed in the “therapy at the time of sample” category. We have now added a new category called “no active therapy” and another explanatory text passage for Table 2.
2. „From the table 2, the author stated that FLT3 expression have PFS impact, but there is no patient numbers on each group.“
Response: We elaborated Figure 3 and Table 2. In addition to the pre-existing Kaplan-Meier analysis (Figure 3B, section 2.2), we have now added a new table (Table 3) which contains the median progression free survival of FLT3-high and FLT3-low patients and the respective p value, as proposed by the reviewer. In this table we also listed the differences between the two groups in regard to ß2-microglobulin levels and ISS stages. This presentation gives a more structured overview over the FLT3 data.
3. „From Table 2, the authors stated that FES score had impact on PFS. Since the risks of survival is totally different in different subgroups, such as good survival of MGUS, the authors should not take all patients from 3 different groups together to see the impact. They should see the impact of FES score of FLT3 in different groups separately.“
Response: We agree that the MGUS/SMM cohort should not be included in the survival analyses, as their prognosis is undoubtedly better and would therefore, bias the results. Indeed, we did not include them in our Kaplan-Meier procedure in the first place, albeit we missed explicitly pointing this out and can see how this caused confusion. Consequently, we have now added a paragraph explaining that the MGUS/SMM cohort was not part of our statistical analyses in section 2.2. Moreover, we excluded the patients, where we were not able to collect substantial follow-up data for further statistical analyses. Thus, all statistical data presented in the results sections 2.3 and below now refer to the 32 patients that we used in our survival analyses. To ensure that our results were not going to be skewed by combining the NDMM and RRMM groups, we compared their PFS in a Kaplan-Meier curve.
4. „The same problem as table 1 in table 2, the number on the table does not match the total patient number in FES low/high groups. Especially the number of therapeutic lines and therapy types did not match the total number in each group. Since the therapy lines and treatment agents had strong impact on patient survival, the authors should make the number in the table right for a clear picture.
Response: As mentioned in point 3, we only included 32 NDMM/RRMM patients in further statistical analyses. Information on their clinical parameters are listed in table 2 and the new Table 3.
5. „From figure 5, how about the expression of normal or MGUS samples?“
Response: Unfortunately, we do not standardly obtain bone marrow aspirates from normal patients and MGUS patients at our Medical Center. Therefore, the CD138 purified cells stem from a collective of bortezomib-sensitive and bortezomib-refractory patients which we published recently (Borjan et al. Frontiers in Oncology Front Oncol 2020 Jan 22;9:1530. doi: 10.3389/fonc.2019.01530. eCollection 2019).
6. „In figure 6, the FLT3 inhibitors, mido and gil can inhibit DNA synthesis in myeloma cell lines in very concentrations, how about the FLT3 expression on these cell lines? Could the authors use other myeloma cell lines with low FLT3 expression?
Response: We thank the reviewer for this remark and added these data. The new Figure 5B contains the FLT3 gene expression values of MM and AML cell lines that were used for proliferation analysis in Figure 6. Interestingly, MM cell lines with high FLT3 expression, such as KMS12-PE and KMS12-BM were more sensitive to the FLT3 Inhibitor gilteritinib than the MM cell line NCI-H929 displaying low FLT3 transcript amounts. These data are discussed in the revised section 2.4.
7. „Some patient’s samples showed potential effects of FLT3 inhibitors, how about the FLT3 expression in these samples? Do these patients belong to high expression group? If so, how about other patients?
Response: As suggested by the reviewer, we performed a FLT3 gene expression analysis on CD138 purified MM cells from patients analyzed in the FLT3 inhibitor apoptosis assays. These data have been incorporated in the new Figure 7. MM#258 had the highest FLT3 expression. MM cells of this patient were also the most sensitive to gilteritinib-induced apoptosis. Due to the few samples analyzed in primary bone marrow apoptosis assays (n=6) we cannot provide a statistical analysis about FLT3 gene expression status and sensitivity to gilteritinib treatment.
Reviewer 3 Report
Normann Steiner and colleagues provided interesting facts about the role of FLT3 in myeloma cells to provide as possible clonal driver novel biomarkers, identify possible underlying mechanisms of development of MM and advance potential therapeutic targets.
However, there are some major points that deserve to be improved before proceeding with the publication:
- The Kaplan-Meier analysis clearly shows the prognostic impact of the considered variable and patients’ prognosis. Did the authors check for a statistical association between FLT3 score and PFS, corrected for the other variables (i.e. ISS, R-ISS, Genetic risk, age, Renal function, Hb, Extramedullary disease - EMD) with significant impact within a uni- and multivariate analysis? A further suggestion in case of positive association might be the Chi square Pearson test and the Camér’s V test (between 0 and 1. 0=independence assumption; 1=maximal dependence). Those are fundamental details and information in this regard should be added to the results. This might be beyond the scope of the manuscript, but I would be interested in the authors ‘comment in this regard.
- The experimental tests performed may be statistically significant but biologically less relevant if placed into a more complex context, such as a statistically powered prospective study. To compensate for these limits, the multivariate Cox’s proportional hazard regression models is a worth tool. Nevertheless, a mandatory assumption needs to be taken into account in order to apply such a model: hazards proportionality. This assumption has to be made in order to proceed with Cox model. If the If the answer is affirmative, this should be better highlighted in materials and methods. If it is not, it is necessary to motivate and discuss the use of alternative models or at least, pinpoint that this might be a study weakness.
- Moreover, authors collected data from a retrospective cohort. Therefore, it would be useful to better clarify the methodological statistical approach used in order to limit, and even avoid statistical biases. I would strongly recommend taking into account those interesting networks.
- Please add exact definition of FLT3 Score </-3 in “Statistical Analysis” section of “Methods”. Please include a figure visualizing absolute FLT3expression across the entire cohort.
- Do original clinical data exist about the translational relevance of the mentioned result? If yes, these elements should be presented, at least in the form of discussion and/or additional figure from a short literature meta-analysis. In discussion section the authors state “Internal tandem duplication (ITD), which leads to autonomous FLT3 activity, occur in approximately 20% of AML patients [26-27].” by alluding to the role of Tandem-duplicated Flt3 constitutively activating STAT5 and MAP kinase and introducing autonomous cell growth in IL-3-dependent cell line, appropriately quoting some seminal manuscript on this regard. Nonetheless, while considering the authors’ analysis results, one suggestion might be the role of recent evidence in this regard, dealing MAPK dependency and its potential in modulation in MM progression and deserving a theragnostic role. Indeed, especially in high risk patients and/or extramedullary myeloma the authors themselves alluded to the pioneering work from Lagana’ et al. who created computational platform for personalized treatment recommendations for RRMM, dissecting also a MAPK dependency. Those are fundamental information in order to deeper validate the evidences discussed in the manuscripts. Nonetheless, this reviewer personally miss an important piece of evidence Nonetheless, extramedullary (EMD) manifestation and brain RR phenotype might deserv a short mention with its own dignity, in light of the above mentioned connection between MAPK, FLT-3 via NRAS/RAS (PMID: 32043788) and prompts curiosity about the potential role of FLT3 while CNS/EMD involvement. Do the authors have data regarding this specific clinical challenge? In the frame of this thinking, the authors introduced and discussed the role of the bone marrow microenvironment supporting tumor progression [reff 12,13]. The above mention published manuscripts (12, 13, PMID: 32043788) and several others uncovered the adhesion and MICROENVIRONMENT system to be pivotal as these stromall-multiple myeloma interactions feed into a vicious cycle propagating disease progression, particularly the EMD relapse.
- There are some stylistic and linguistic gleanings that require a careful revision, such as an images quality optimization a professional linguistic editing.
Author Response
1. “The Kaplan-Meier analysis clearly shows the prognostic impact of the considered variable and patients’ prognosis. Did the authors check for a statistical association between FLT3 score and PFS, corrected for the other variables (i.e. ISS, R-ISS, Genetic risk, age, Renal function, Hb, Extramedullary disease - EMD) with significant impact within a uni- and multivariate analysis? A further suggestion in case of positive association might be the Chi square Pearson test and the Camér’s V test (between 0 and 1. 0=independence assumption; 1=maximal dependence). Those are fundamental details and information in this regard should be added to the results. This might be beyond the scope of the manuscript, but I would be interested in the authors ‘comment in this regard.”
Response: We thank the reviewer for giving us the opportunity to address this issue. As outlined in Table 1 of the manuscript, we were able to include 42 patients in this pilot study. In this situation, a multivariable analysis or a multivariable Cox-Regression is questionable due to the lack of power and progressed cases. Nevertheless, we have checked potential associations of important predictors with PFS by univariate KM analyses and potential associations between the predictors using the CHI²-test. The methodological information was added in the statistical methods section. To improve the readers understanding, we extended the information in Figure 3 by results of a univariate Cox-Modell (Case-Numbers, Hazard-Ratio, P-Value) and show these data the new Table 3.
2. „The experimental tests performed may be statistically significant but biologically less relevant if placed into a more complex context, such as a statistically powered prospective study. To compensate for these limits, the multivariate Cox’s proportional hazard regression models is a worth tool. Nevertheless, a mandatory assumption needs to be taken into account in order to apply such a model: hazards proportionality. This assumption has to be made in order to proceed with Cox model. If the If the answer is affirmative, this should be better highlighted in materials and methods. If it is not, it is necessary to motivate and discuss the use of alternative models or at least, pinpoint that this might be a study weakness.“
Response: Please see our response to Question 1.The PH-assumption for predictors were checked by inspection of the KM-curves.
This information was added in the statistical methods section (please see response Q1)
3. „Moreover, authors collected data from a retrospective cohort. Therefore, it would be useful to better clarify the methodological statistical approach used in order to limit, and even avoid statistical biases. I would strongly recommend taking into account those interesting networks.“
Response: We thank the reviewer strongly for his comment. Following the reviewers suggestion, we inspected several potential associations of predictors or confounders, but we did not find relevant biases. Nevertheless, we also added a limitation remark in the discussion.
4. „Please add exact definition of FLT3 Score </-3 in “Statistical Analysis” section of “Methods”. Please include a figure visualizing absolute FLT3 expression across the entire cohort.“
Response: Thank you for this suggestion. We have added a detailed definition of the FES (FLT3 expression score) in the “statistical analysis” section 4.11 and hope that our description leaves no questions unanswered. We have also created an additional figure (Figure 3A) visualizing the FES (FLT3 expression score) distribution over the three patient groups (MUGS/SMM vs. NDMM vs. RRMM).
5. „Do original clinical data exist about the translational relevance of the mentioned result? If yes, these elements should be presented, at least in the form of discussion and/or additional figure from a short literature meta-analysis. In discussion section the authors state “Internal tandem duplication (ITD), which leads to autonomous FLT3 activity, occur in approximately 20% of AML patients [26-27].” by alluding to the role of Tandem-duplicated Flt3 constitutively activating STAT5 and MAP kinase and introducing autonomous cell growth in IL-3-dependent cell line, appropriately quoting some seminal manuscript on this regard. Nonetheless, while considering the authors’ analysis results, one suggestion might be the role of recent evidence in this regard, dealing MAPK dependency and its potential in modulation in MM progression and deserving a theragnostic role. Indeed, especially in high risk patients and/or extramedullary myeloma the authors themselves alluded to the pioneering work from Lagana’ et al. who created computational platform for personalized treatment recommendations for RRMM, dissecting also a MAPK dependency. Those are fundamental information in order to deeper validate the evidences discussed in the manuscripts. Nonetheless, this reviewer personally miss an important piece of evidence Nonetheless, extramedullary (EMD) manifestation and brain RR phenotype might deserv a short mention with its own dignity, in light of the above mentioned connection between MAPK, FLT-3 via NRAS/RAS (PMID: 32043788) and prompts curiosity about the potential role of FLT3 while CNS/EMD involvement. Do the authors have data regarding this specific clinical challenge? In the frame of this thinking, the authors introduced and discussed the role of the bone marrow microenvironment supporting tumor progression [reff 12,13]. The above mention published manuscripts (12, 13, PMID: 32043788) and several others uncovered the adhesion and MICROENVIRONMENT system to be pivotal as these stromall-multiple myeloma interactions feed into a vicious cycle propagating disease progression, particularly the EMD relapse.“
Response: To our knowledge, there is no clinical data available on flt3 inhibition in multiple myeloma so far. Although we analyzed several RRMM patients our patient cohort did not contain patients with extramedullary disease. So we cannot add data/analysis in this regard. However, we now added to the discussion section3 molecular pathways affected by FLT3 signaling. Pathways like PI3K/AKT1 or BRAF/MEK can be used for targeted therapy approaches in MM and the above suggested paper is now referenced. It is tempting to speculate that combined therapies including flt3 inhibitors might especially useful in the setting of extramedullary disease and we will take this into account in future studies. Thank you for drawing our attention to this.
6. „There are some stylistic and linguistic gleanings that require a careful revision, such as an images quality optimization a professional linguistic editing.“
Response: Image quality has been improved (Figure 1). Moreover, most figures have been elaborated optically and contentwise to grant readers a better understanding of the presented data. The manuscript, in particular the discussion part, has been rewritten for stylistic and linguistic gleanings. The manuscript has been carefully proofread.
Round 2
Reviewer 1 Report
The author has fully addressed my comments.
Reviewer 2 Report
The authors modified the manuscript and the studies according to the suggestions. Therefore, I agree to publish in the journal.
Reviewer 3 Report
The authors have clarified several of the questions I raised in my previous review. Most of the major problems have been addressed by this revision. No further comments from this reviewer.